# POSITION: WE SHOULD READ AND WRITE PAPERS DIFFERENTLY IN THE AGE OF AI

## ABSTRACT

Research papers follow conventions that are widely agreed on and enforced by the scientific community. The established paradigm is that each paper is self-contained, including contextualization of the proposed method. This static author-side contextualization creates several risks, such as incentives to promote a certain biased narrative or the outdatedness of related literature. Furthermore, readers might have to rely on external resources to assist their understanding if they have a substantially different background from the authors. At the same time, the amount of research output is rapidly increasing in tandem with the usage of AI tools for interacting with papers. We argue that the age of AI enables a paradigm shift in which research dissemination transitions to a more reader-centered approach, and that this shift should be approached proactively by the scientific community.

## 1 INTRODUCTION

The process of research dissemination via scientific papers is well established and has remained stable for many years. Every paper is expected to explain its motivation and contribution, be technically self-contained, backed up by empirical evidence or theoretical proofs, and embed itself in the existing literature. This structure[1] has proven effective; however, it has several downsides. Contextualization of the novel contribution is written from the perspective of the authors. Apart from the risk of inaccuracies, authors have an incentive to omit findings that oppose their narrative. Since the contextualization is statically included in the paper, noise and biases compound over time. Additionally, papers can only argue based on the state of knowledge at the time of writing, which might have since changed. Lastly, the conventional paper structure is created for human readers, often including redundancies to aid understanding. At the same time, static contextualization can be challenging to comprehend for readers with substantially different background from the authors. The advent of complex large language models (LLMs) and their human-like level of natural language understanding, i.e., the age of AI[2], presents the opportunity to address the above issues.

### 1.1 SUMMARY OF OUR POSITION

We argue for the transition to a paradigm that assumes paper interaction to be ubiquitously mediated by AI. The core targets of our approach are to make paper writing more efficient, reduce author-induced biases, and make paper reception more effective. Our proposal is to switch from a provided-by-author to an AI-generated-for-reader strategy. In this vision, papers are strictly reduced to the description of novel contributions, i.e., methodology, its formalization, and evaluation, the so-called *core paper*. Any additional context, such as motivation, technical background, and related literature sections, is generated for the reader by AI, tailored to their specific knowledge and intentions. We call this *immersive reception*.

Further, the traditional role of survey papers – providing a structured overview of a field of research – will be taken over by reader-directed LLMs. We argue instead for the introduction of two new mechanisms: (1) continuous AI-generated *state-of-the-field reports* provided by scientific societies representing the state of knowledge in real-time, and (2) *cornerstone papers* serving as long-term quality safeguards through an extensive reviewing process.

### 1.2 ASSUMPTIONS

Before presenting our proposals in more detail, let us first introduce the assumptions on which we operate.

---

[1]Our examples and argumentation are focused on the areas of computer science and computer engineering. However, the presented ideas apply to the entire scientific community in principle.

[2]We refer to AI and AI agents as a collective term for LLM-based tools of arbitrary complexity and autonomy.

**Scientific research will require human guidance for the foreseeable future.** Our core assumption is that human creativity and cognition will remain the guiding centerpiece of scientific progress for the foreseeable future. If scientific research were to become entirely autonomous, this paper would be pointless. Accordingly, we argue for redesigning artifacts and processes of communicating science such that they improve interactions with AI and humans likewise.

**AI will only get better.** While AI is already increasingly used in the writing, reviewing, and reception of scientific papers (Kacena et al., 2024a; Hu, 2024), current tools often reach their limits with complex technical nuances. While some proposals in this manuscript might not be realizable with robust quality at this point in time, we believe they will be in the next few years. However, we base our core argumentation mainly on what we deem possible with the tools available today, clearly marking ideas predicated on future improvements in AI capability.

**The plurality of AI vendors will ensure impartiality.** Our argumentation relies in part on the assumption that AI is generally more objective than individual researchers. We recognize the risk of AI tools being controlled by economic actors with potential incentives to introduce partiality toward certain ideas. However, we assume (1) a sufficient degree of exchangeability such that no AI vendor would rationally decrease output quality for the sake of a specific agenda, and (2) a sufficient plurality of tools in use by the global scientific community such that biases cancel out overall.

## 2 Papers in the Age of AI

Papers are the foundational mechanism of communicating and disseminating scientific progress. There are several general stages of interaction, serving different purposes: First, a researcher would search for related works, obtaining an overview. Second, promising papers are read in more detail to understand if they are relevant. They might then be discussed as related literature or provide arguments for a certain storyline. Third, the reader might decide to replicate results, use, or build upon a previously proposed method. For this, it is important that the paper is reproducible. Our vision for papers in the age of AI is guided by two observations: Different paper sections serve different purposes as readers progress through stages of interaction, from discovery to deep engagement, and early stages are particularly well-suited for being automated via AI agents.

Additionally, readers have diverse needs and preferences to obtain scientific knowledge (Lee, 2003), and we believe that immersive reception enables them to fulfill this need. We want to stress that we do not claim that our proposed framework will be the dominant way of writing papers in the future, but rather open the discussion by showing alternatives to the current format. Such alternative formats have become more prominent in recent years, e.g., arXiv also serves an HTML version of the conventional PDF to enhance accessibility and provide a responsive experience (Frankston et al., 2024).

In this section, we describe both the perspective of the authors and the readers under this new paradigm of immersive reception. Additionally, we discuss the role of survey papers and the implications on citation scores. We conclude by suggesting a way forward to adopt our proposed system.

### 2.1 The Paper of the Future

This section is concerned with the perspective of the authors. Many elements of papers today are repetitions and reformulations serving as guidance for human readers. This is unnecessary under the paradigm of immersive reception, in which a core paper only needs to provide truly novel information that could not be inferred and generated by AI. This increases the efficiency of writing papers while simultaneously reducing the risk of author-induced biases in the contextualization.

We specifically discuss each conventional paper section and argue whether it is still relevant or how it should change in the future. We want to emphasize that the underlying argumentation could be realized in different ways, and the presented structure is only exemplary. A concrete example is provided in Appendix A.2.

**Abstract and Paper Summaries** The abstract is a concise summary of the paper. It does not introduce new information, but rather provides framing. Accordingly, it is not part of a core paper and can rather be generated by an AI agent tailored for the reader. This would also directly avoid the need for lay summaries (Goldstein & Krukowski, 2023), which summarize papers in plain language. These summaries were introduced in major ML conferences in recent years[3] in an effort to better communicate the impact of a paper to a broader audience. This precisely aligns with our call for an immersive reading experience.

---

[3]E.g., ICML'25: https://icml.cc/Conferences/2025/CallForPapers

**Introduction**   This section is partially part of a core paper. The introduction section is written to attract a very broad audience towards the paper at hand. It is often composed of an introduction to the problem setting, a more technical motivation, the contributions of the paper, and an overview of the organization of the paper, which can mostly be dynamically generated by an AI agent. In particular, the AI agent can emphasize why the paper is relevant to the reader, instead of singularly reflecting the perspective of the authors. The only element still present in a core paper is a technical motivation. This motivation details why existing solutions are insufficient and precisely formulates the research gap or research questions the paper is addressing.

**Related Work**   This section is not part of a core paper. While an overview of related literature can be helpful for human readers, it is virtually impossible to provide a comprehensive list. Accordingly, authors have an incentive to provide a selection supporting a specific narrative that is favorable to their views and contributions. Shifting to a reader-generated paradigm avoids this source of bias and can increase relevance for the individual background and research questions of readers. Furthermore, static related work sections are, counting the time from initial publication to reception, often already outdated. In contrast, readers might want to read one comprehensive up-to-date literature review while exploring a new topic, and are usually not interested in the progressive updates in each paper. We note that removing the related work section heavily interferes with citation scores. We discuss the resulting implications in more detail in Section 2.4.

**Background**   This section is part of a core paper, albeit in a more concise form. The background section does not have a universally agreed-upon structure but is commonly used to introduce notation and technical building blocks, as well as to formalize the problem statement. Notational clarity becomes more important as the amount of explanation throughout the paper is reduced. While technical building blocks need to be introduced and cited, they do not need to be explained in detail insofar as they follow a standard notation. Authors can essentially assume that the reader-side AI agent can explain any required context and focus on formalization. The problem statement is an essential part of a core paper and should be formalized as much as possible to avoid any ambiguity. It is clearly marked, and AI agents are instructed to rely on it to determine the scope of a paper.

**Methodology**   This section is a central element of a core paper, and generally follows the established format. Just as with the background section, the amount of explanation can be reduced since the reader-side AI can provide customized guidance during immersive reception. We again want to stress the proper formalization of the approach for the AI agent to correctly guide the reviewer to the approach, possibly with the help of a running example, and contrast the approach to related work. While automated theorem provers (Nipkow et al., 2002; de Moura et al., 2015; Slind & Norrish, 2008) can help with the formalization, this might not always be applicable for all paper formats.

**Evaluation & Experimental Results**   If experimental validation is part of the research objective, this section is an integral part of a core paper. Papers will increasingly be compared automatically by AI agents, necessitating a more structured approach to reporting results. Venues or entire scientific communities should establish standards, constituting a hard criterion for acceptance. An example of such an evaluation framework for deep reinforcement learning is presented in (Agarwal et al., 2021). We discuss the topic of reproducibility in more detail in Section 2.3.

**Limitations**   LLMs tend to overgeneralize and be overly optimistic (Dahlgren Lindström et al., 2025; Peters & Chin-Yee, 2025), which is why a special focus should be placed on a comprehensive discussion of limitations as part of a core paper. While the insufficient self-reporting of limitations is an issue already today (Lago et al., 2024), this would be substantially exacerbated by the use of AI. This section is a priority of any core paper, and reviewers would be asked to ensure that the implications of all relevant limitations are appropriately examined (Lago et al., 2024; Brutus et al., 2013).

**Conclusion**   Similar to the abstract, this section does not contain any new information and is, therefore, not part of a core paper.

In summary, our proposed core paper begins with a technical motivation, followed by the introduction of notation, the formalization of required background, and a precise problem statement. Afterwards, the methodology is described. Finally, results from empirical experiments are reported, closing with a dedicated discussion of limitations. To obtain a glimpse of the papers of the future, we highly recommend checking the examples in Appendix A.

### 2.2   IMMERSIVE RECEPTION

We have now established how authors could write papers in the age of AI. In this section, we want to detail the implications for the reader. We want to stress that this marks a paradigm shift in scientific paper perception. Conventionally

written papers inherently mirror the perspective of the authors, who have an incentive to oversell their work. Additionally, this perspective is static from a certain time in the past, carrying the risk of perpetuating assumptions that are meanwhile disproven and compounding inaccuracies through incorrect citations. In contrast, our new proposed way to write papers is tailored to the reader, enabling them to have an immersive reception based on the current state of knowledge.

We envision a framework where the AI agent is aware of the experience, past projects, and new ideas of the reader. Using this knowledge, the AI agent can dynamically adjust the content of a paper. Here, the possibilities are manifold, and we exemplarily list the advantages of the immersive reception for the reader below and accompany each advantage with a corresponding example in Appendix A.

**Minimize the spiel, maximize the impact.** Researchers are often overloaded with work, leaving them little to no time to read papers that might impact their work. Our envisioned framework would allow experienced researchers to quickly read through the main content of the paper. While they could already try to directly skip to the relevant place in conventional papers, they can find themselves going back and forth in the paper as they read into the details. An AI agent could circumvent this problem, as it can summarize the relevant details to the experienced researcher, while giving a full introduction to new researchers in the field (Appendix A.1).

**Diving deeper into a paper.** As the reader wants to dive deeper into the paper, the AI agent can dynamically provide the relevant details until the reader obtained the background to read and understand the core paper. This process can also be personalized based on preference. For example, some readers like to be aware of all the background required before reading the novelties of the specific approach, while others like to only get exposed to background information when it is required to understand the next portion of the paper (Appendix A.3). Furthermore, examples can be rendered interactive and, if desired, accompanied by questions to the reader to foster understanding.

**Re-surfacing old memories.** Core to an immersive reading reception is the ability to dynamically expand on the paper to provide additional context for the reader. While the AI agent is aware of the reader's experience, it can still happen that the reader needs to refresh their memory on a certain topic. The AI agent can then, for example, bring up a connection to another project where the reader already worked on the problem at hand, such that the reader can quickly re-familiarize themselves with the topic, and in addition, obtain a uniquely personalized connection that is not possible without our proposed immersive reception (Appendix A.4).

**Off to new adventures.** Such unique connections can bring a reader quickly into uncharted territories, and the AI agent can help sail them safely. Exploring unique connections, possibly obtained through a flash of inspiration, is not uncommon while reading another work, and we believe that following such instincts is an integral part of the scientific process. Our proposed immersive reading experience enables the AI agent to make the reader aware of the opportunities while warning them of potential pitfalls, all provided with the relevant literature to help overcome them (Appendix A.5).

**No need to learn new notation.** Many papers, in particular theoretical papers, lean heavily on mathematical notation to describe their concepts and prove their findings. However, different fields, subfields, and even research labs have different conventions for their notation, such that it can be difficult to understand the results and follow their proofs. This becomes even more apparent when a reader just skims over many papers in parallel to compare existing approaches, each introducing their own notation. With the formalization of the background and methodology sections in the core paper, an AI agent could harmonize these notations the way the reader is used to them, drastically reducing the unnecessary burden to learn a new notation for each paper (Appendix A.6).

**Bringing down knowledge barriers.** While conventionally written papers introduce a reader to the topic, they often still require a certain level of knowledge to follow along. This hinders researchers from different domains from accessing the knowledge. While lay summaries (Goldstein & Krukowski, 2023) can help overcome this barrier, a single paragraph in plain language cannot provide for the diverse needs of the audience.

## 2.3 THE NEW ROLE(S) OF SURVEY PAPERS

Survey papers contain a review of a specific area of research, explaining and comparing methods, and often providing overarching structural or theoretical insights. They serve as anchor and low-barrier entry points for literature search. There is a significant uptick in published artifacts already today, and this will only continue to increase (Chu & Evans, 2021; Piller & Grech, 2021; Brandt & Tague, 2023). This trend will likely result in the traditional role of survey

papers largely being taken over by reader-side AI (Alshami et al., 2024), in turn rendering long-term quality control even more important. To facilitate this quality control, we envision the three mechanisms outlined below.

**Reproducibility Papers**   Empirical validation is essential for novel approaches in almost all domains. Reproducibility of results is a perpetual challenge (Martínez-Cajas et al., 2024; Hayot-Sasson et al., 2025; Stodden et al., 2018; National Academies of Sciences, Engineering, and Medicine, 2019) and, because setup and implementation are complex and nuanced, is not likely to be automated by AI (National Academies of Sciences, Engineering, and Medicine, 2019; Chan et al., 2024; Ziemann et al., 2023). While dedicated reproducibility papers have been published independently (Heil et al., 2021; Cavenaghi et al., 2023) and in the context of so-called reproducibility challenges (Pineau et al., 2021; 2019), they are still notably sparse. A central obstacle is establishing appropriate incentives for investigating reproducibility. The introduction of state-of-the-field reports, consequently citing reproducibility studies, could alleviate this issue. Another approach could be to introduce *reciprocal reproducibility*, i.e., venues demanding a reproducibility result for every submitted paper. Reader-side AI would generally be instructed to be skeptical of results until they have been sufficiently reproduced.

**Continuous State-of-the-Field Reports**   AI enables the rapid summary and structuring of hundreds of papers at once (Khraisha et al., 2024). At the same time, a prompt written by someone with some knowledge of the field will lead to a more precise result. Leveraging these observations, we propose that large venues and scientific societies take responsibility for hosting AI-generated state-of-the-field reports, representing the current research landscape in quasi-real-time. These reports put an emphasis on reproducibility results and can serve as a reliable entry point for researchers new to a certain field and as a first quality gate. This approach builds upon the concept of living systematic reviews, which provide continuously updated evidence synthesis (Elliott et al., 2014; 2017; Synnot et al., 2021; Breuer et al., 2023).

**Cornerstone Papers**   In parallel to the speed-up of paper production and dissemination, the risk of inaccurate or wrong information proliferating increases (Chu & Evans, 2021; Piller & Grech, 2021). A possible instrument to mitigate this is to establish heavily quality-controlled cornerstone papers, which act as long-term points of truth. This essentially institutionalizes the concept of scientific curation (Plon et al., 2025; Bugbee et al., 2023; Govoni et al., 2019). Cornerstone papers are ideally based on a recent state-of-the-field report and carefully manually edited by several high-profile experts in the field. The topics can be proposed by the community, but the editorial team is based on invitations to ensure strong commitment and a high level of prestige.

## 2.4 IMPACT ON CITATION SCORES

Citation scores are used as an initial estimate of whether a paper was adopted by the broad research community. Our proposed paradigm shift in writing papers substantially influences the number of citations a paper gets. By removing large portions of the introduction and related work, only papers that a new paper directly builds upon get cited as part of their core submission. This will reinforce the role of survey papers to maintain an overview and avoid parallel threads of research.

However, only counting citations through the core paper no longer paints the full picture of the impact of a paper. The immersive reception would allow for an additional, more dynamic citation score: the receptive citation score. In this receptive scoring system, citations are collected whenever a reader requires a paper to understand another paper, rewarding fundamental works to bring inexperienced readers to the topic, and closely related work to understand details.

Additionally, the receptive citation score also has a secondary effect: It measures where a scientific community lacks experience, and additional contextual information is requested during the immersive reception. In particular, looking at the receptive citation score on a temporal dimension enables valuable insights: We would expect such a citation score to initially rise when an approach gains interest in a research community, but fall off as another approach becomes dominant. In contrast, the conventional system would always have the first approach with a higher score, as it starts earlier to collect scores, and later both collectively collect citations through the related work section. If, however, the approach remains mainstream, the receptive citation score will remain high as new researchers come into the field and want to understand the approach.

## 2.5 THE WAY FORWARD

The core challenge for transitioning toward the outlined vision of immersive reception is the development of suitable tools and platforms. The most likely stakeholders to drive this change are the already established points of paper

interaction, such as search platforms and literature management software. Since the inclusion of AI into such tools would signify a substantial competitive advantage, the economic incentives are strong.

The role of the scientific community will be to provide source material that is well-suited for AI-based interaction, with minimal bias and long-term quality assurance. As long as our proposed paradigm is not mainstream, venues could request authors to make a dual submission with one conventional version and one core paper version. This would initially be more work for authors, but it would serve as a risk-free path to validating the concept of immersive reception in the near-term future.

Scientific societies will have to take on more responsibility for safeguarding and structuring the scientific process. This is mainly due to the rapidly increasing volume of research output, which is, in part, facilitated by the use of AI (Kacena et al., 2024a; Hu, 2024). The landscape of community organization is too complex to provide specific suggestions for implementation. However, a clear universal recommendation is to take on this transition proactively by facilitating public discussions and establishing task forces.

Nevertheless, we acknowledge that some practical challenges need to be addressed when implementing such a framework. For example, the abstract is also often displayed on a static website where the immersive framework is not available. Licensing of publishers might hinder an AI agent from accessing a paper, resulting in a bias towards open-access publications and preprint servers. Standards for authenticated delegation (South et al., 2025) would need to be implemented to robustly interact with closed-access sources. Further, an AI agent might illustrate a running example by translating pseudo-code into runnable code, executing it with the right data, and returning results – along with traceable steps – requiring enough formalization to minimize errors.

## 3 ALTERNATIVE VIEWS

### 3.1 BUT AI HALLUCINATES!

AI systems are prone to generating incorrect or misleading information. This risk is real and well-documented. One can argue that relying on AI to generate critical parts of the context of a paper risks introducing inaccuracies at scale. Unlike human-written text, which undergoes peer review and revision, AI outputs can vary with prompts, models, or vendors, and may misrepresent related work or background information. Even small inaccuracies could mislead readers, distort the perceived state of the field, or propagate false claims if unchecked.

**Counter:** While the risk of hallucinations is real, it is not unique to AI. Human authors also make mistakes, sometimes systematically due to bias or incentives to oversell their contributions. Unlike static human-written text, AI-generated content can be flagged and corrected in real time as an AI agent becomes aware of other papers pointing out flaws, reducing the persistence of errors. We have to expect that authors will increasingly deploy AI agents to write the contextualization for their papers in the future. Naturally, these have an incentive to oversell the work, such that factually false statements, biased framing, and misleading comments are likely to increase if we leave the system unchanged. In contrast, reader-side AI agents have no vested interest in exaggerating the novelty or impact of a particular paper. Crucially, if enough diversity of AI agents exists, readers can choose AI agents that best represent the scientistic discourse and bring papers into context. As models improve, reader-side AI agents will reliably contextualize papers in the scientific discussion, and by freeing authors from repetitive contextual writing, authors can devote more effort to rigorous formalization – ultimately improving paper quality.

### 3.2 SPENDING TIME ON THE PAPER IMPROVES QUALITY

The process of constructing a coherent narrative, from motivation to conclusion, requires organization, reflection, and synthesis that strengthen the research itself. If authors restrict themselves to drafting only a minimal core paper, they risk producing fragmented or underdeveloped contributions. The act of composing these sections forces authors to engage deeply with prior literature, reflect on how their contribution fits into the broader field, and critically assess both originality and limitations. Without this process, authors may remain unaware of important related work or overestimate the novelty of their results.

**Counter:** In practice, many authors already treat introductions and related work as formulaic, often repeating standard framings or citing selectively to strengthen their own narrative. Offloading this to reader-side AI avoids entrenched biases and ensures that contextualization is always current rather than frozen at the moment of publication. With our proposal, authors are still required to provide precise motivation, well-formalized methodology, background, and evaluation. We argue that writing precisely these sections requires the most reflection for a coherent argument,

which will still persist under the new format. Additionally, the core papers will still undergo peer-review processes, such that works that are closely related to existing literature can still be flagged.

### 3.3 Immersive Reception is Inefficient

While current reports place the energy consumption of generating a single LLM response at well below 1 Wh, reasoning models and agentic pipelines can increase this by several orders of magnitude (Elsworth et al., 2025; Jegham et al., 2025). For immersive reception, contextualization is created for every individual reader instead of only once by the author. This could increase the energy impact of popular papers by a factor of 10,000 or more.

**Counter:** Making the contextualization more relevant to the reader makes reception more effective. This increase in accessibility reduces the number of LLM interactions and search time, overall accelerating research. Since the relative environmental impact of LLMs is already substantially smaller than that of human labor (Ren et al., 2024), an acceleration of the scientific progress would decrease its environmental impact in the long term. In the short term, platforms offering immersive reception could cache contextualization for common user profiles, e.g., different levels of expertise, to avoid redundant computation. Considering the likely scenario that users will use reasoning models to better understand papers anyway, a structured platform-driven approach might even prove more efficient.

### 3.4 You can already let AI write a personalized intro

One might argue that our proposal is unnecessary because existing AI tools can already generate personalized summaries or introductions for any paper. Readers today can upload a PDF into a chatbot or use browser extensions to obtain a custom introduction tailored to their background. Why, then, should the entire structure of papers be changed?

**Counter:** While technically true, ad hoc AI usage is fragmented, inconsistent, and depends heavily on the prompting skills of the reader. Our proposal envisions immersive reception as a systematic, community-driven framework rather than a patchwork of individual hacks. Moreover, to avoid the downsides of author-generated contextualization, AI tools would have to work around it. In contrast, when preparing a core paper, authors can focus on the novel parts of their paper with high information density instead of writing and polishing text that is then largely disregarded. This increases the efficiency of the scientific process while reducing the risk of compounding and long-lasting noise and biases.

### 3.5 Core Papers Counteract Long-Term Archival

The process of writing papers has proven to be effective for centuries, providing long-term archival of scientific progress. These papers are written in formats that will be accessible well beyond any trend. Additionally, as the papers are self-contained and peer-reviewed, they do not miss on contextualization even if related work is forcefully destroyed or becomes otherwise unavailable. By contrast, AI-mediated contextualization depends on ephemeral models and platforms that may become obsolete, undermining the long-term preservation of knowledge.

**Counter:** We agree that our proposal is a paradigm shift from the conventional way of writing papers. Thus, we argue in Section 2.5 for dual submissions until our proposed format has proven effective. We also expect that this new format will require several iterations until it converges to a new dominant way of writing papers, minimizing the amount of time authors have to write repetitive contextualization while providing sufficient and sufficiently unambiguous details for AI agents to contextualize it for readers. If the scientific community agrees that self-contained papers should be preserved for long-term archival, an extended version representing the view of the authors could be stored alongside the core paper.

## 4 Related Work

**AI in scholarly writing and disclosure.** Studies and editorials note real efficiency gains alongside frequent inaccuracies and fabricated citations, underscoring the need for human oversight and transparent reporting of AI use (Kacena et al., 2024b; Khalifa & Albadawy, 2024; Khlaif et al., 2023). Complementing this, "AI Usage Cards" propose a concrete, standardized disclosure framework for transparency, integrity, and accountability (Wahle et al., 2023).

**Machine-readable and interactive papers.** Prior work has pushed papers beyond static PDFs toward formats friendlier to both humans and machines: ArXiv's recent HTML initiative argues HTML is the most impactful step for

accessibility and downstream machine processing (Frankston et al., 2024), which also enables more interactive and explorable articles as demonstrated on Distill (Sanchez-Lengeling et al., 2021). The idea of making papers and supplementary material machine-readable already gained traction before AI became widely adopted (Wilkinson et al., 2016; Bechhofer et al., 2013; Priem et al., 2022), which also requires an efficient way to structure scientific research (Groth et al., 2010; Kuhn & Dumontier, 2015). Similarly, the need for a personalized reading experience has been explored before AI (Dolog et al., 2004; Cachola et al., 2020).

**Dynamic evidence synthesis and "living" overviews.** In medicine and guidelines, "living" evidence models show how continuously updated syntheses can replace static reviews. Living systematic reviews and living guidelines provide timely, rolling updates rather than episodic rewrites – conceptually close to our proposal for continuous state-of-the-field reports maintained by societies and powered by AI summarization (Elliott et al., 2014; Breuer et al., 2023).

**Peer review under AI pressure.** Position papers and recent reporting describe stress on peer review from soaring submissions and LLM use. Investigations show emerging risks like prompt-injection attempts targeting AI-assisted reviews (Lin, 2025).

## 5 CONCLUSION

AI is no longer an optional accessory to scientific writing and reading; it is already reshaping how papers are produced, searched, and consumed. Maintaining the current processes and artifacts of scientific communication risks entrenching inefficiencies and biases that will only grow as research output accelerates. We have argued for a shift from static, author-centered papers to concise core papers combined with immersive, reader-tailored immersive reception. This paradigm holds several promises: reducing redundant contextualization, improving reproducibility through clearer formalization, and enabling dynamic, personalized engagement with scientific knowledge. At the same time, it safeguards long-term archival value through core contributions while leaving contextual framing to adaptive AI agents. This reduces the risk of author-induced biases and facilitates research being embedded in the state of the art long after a paper was written.

Realizing this vision requires coordinated steps from the community. Authors can begin by experimenting with dual submissions, venues can encourage standardized core formats, and societies can invest in quality-control mechanisms such as state-of-the-field reports and cornerstone papers. The way forward is not without challenges, but the alternative – clinging to formats designed for a pre-AI era – is unsustainable. We invite the research community to proactively explore and refine these new models. AI is here to stay, and conducting business as usual is simply not a viable option.

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

## A  FULL EXAMPLE: MERGESORT

To obtain a glimpse of a future paper, we provide several example immersive reading experiences in this section. All immersive reading experiences located in the black boxes have been generated by ChatGPT-5[4] with only minimal editing from our side (e.g., removing introductory sentences outputted by the LLM and formatting). Each prompt included the manuscript of this position paper and a minimal set of instructions to generate the respective example. We want to stress that these examples are just the beginning as AI gets better and one builds dedicated tools for the immersive experience. We restrict ourselves to simple examples that can be conveyed in a paper format.

In this example, we give an immersive reading experience to a hypothetical novel sorting algorithm, MergeSort (Knuth, 1997, Sec. 5.2.4), both tailored to an undergraduate student who has never heard about sorting algorithms and runtime complexities, and to a professor who is fully aware of the literature.

### A.1  IMMERSIVE INTRODUCTION

The immersive introduction for the student is on a very high level, slowly guiding them to the novelties of MergeSort.

---

**☰ Immersive reading experience for an undergraduate student**

Sorting means arranging items (like numbers or words) in a particular order, such as from smallest to largest. This is a fundamental task in computer science because it makes searching, comparing, and analyzing data much easier. For example, online shopping sites use sorting when they show you products from cheapest to most expensive.

In this paper, we look at a method called MergeSort. Imagine having a messy deck of cards: instead of trying to sort the whole deck at once, you split it into smaller piles, sort those piles, and then carefully combine them back into a single sorted deck. MergeSort follows this exact idea — break down, sort, and then merge.

Why is this important? Because MergeSort is faster and more reliable than simple sorting methods you may try yourself (like swapping adjacent items one by one). It gives us a systematic way to handle large collections of data efficiently.

---

In contrast, the immersive introduction for the professor is much more detailed, allowing them to quickly assess the novelties of MergeSort.

---

**☰ Immersive reading experience for a professor**

Existing sorting algorithms such as InsertionSort and SelectionSort exhibit $\Theta(n^2)$ runtime complexity, which is prohibitive for large input sizes.

We introduce MergeSort, a divide-and-conquer algorithm that achieves $\Theta(n \log n)$ runtime in the worst case. The method recursively partitions an array into subarrays, sorts these independently, and merges them in linear time. This approach guarantees predictable performance regardless of input distribution, in contrast to methods whose efficiency degrades on adversarial inputs.

Preliminary evaluations demonstrate that MergeSort outperforms insertion sort by orders of magnitude on large inputs, with performance comparable to the best-known techniques while maintaining deterministic guarantees. Beyond efficiency, MergeSort offers a natural pathway toward parallel execution, as independent recursive calls can be distributed across processors.

We argue that MergeSort establishes a new baseline for comparison in algorithmic efficiency, with implications for both theoretical computer science and practical applications in systems requiring robust large-scale data handling.

---

### A.2  CORE PAPER

Crucially, the authors do not have to provide any high-level introduction to their paper, and they can focus on the contribution of their approach. For example, their submitted manuscript could be just a single page (excluding experiments).

---

[4]For detailed information on the GPT-5 model, refer to `https://cdn.openai.com/gpt-5-system-card.pdf`.

**☰ Core Paper: MergeSort**

## TECHNICAL MOTIVATION

For inputs drawn from a totally ordered domain, elementary comparison-based algorithms (e.g., adjacent-swap schemes) exhibit quadratic time in the worst case. We target a method with (i) worst-case $\Theta(n \log n)$ comparisons, (ii) stability, and (iii) predictable behavior under adversarial input orders, while enabling straightforward parallel execution.

## BACKGROUND

**Model.** We operate in the comparison model on a totally ordered set $(\mathcal{X}, \leq)$. An input is a finite sequence $A = \langle a_1, \ldots, a_n \rangle \in \mathcal{X}^n$. A *stable* sort outputs a sequence where equal elements appear in their original order.

**Correctness specification.** Given $A$, the output $B = \langle b_1, \ldots, b_n \rangle$ must satisfy:

**Sorted:** $b_1 \leq b_2 \leq \cdots \leq b_n,$ (1)

**Stability:** $\forall i < j,\ a_i = a_j \Rightarrow \mathrm{pos}_B(a_i) < \mathrm{pos}_B(a_j).$ (2)

**Performance objective.** Worst-case time $T(n) \in \Theta(n \log n)$; auxiliary space $S(n) \in \Theta(n)$ (or $o(n)$ extra words with in-place merging).

## METHOD

### DIVIDE-AND-CONQUER SCHEME

---

**Algorithm 1** MERGESORT($A$)

---

**Require:** Array $A[1..n]$
1: **if** $n \leq 1$ **then**
2:     **return** $A$
3: **end if**
4: $m \leftarrow \lfloor n/2 \rfloor$
5: $L \leftarrow$ MERGESORT($A[1..m]$)
6: $R \leftarrow$ MERGESORT($A[m+1..n]$)
7: **return** MERGE($L, R$)     ▷ stable linear-time merge

---

**Algorithm 2** MERGE($L, R$)

---

**Require:** Sorted arrays $L[1..\ell]$, $R[1..r]$
1: $i \leftarrow 1$, $j \leftarrow 1$;   create output $B$ of length $\ell+r$
2: **for** $k = 1$ **to** $\ell+r$ **do**
3:     **if** $i \leq \ell$ **and** ($j > r$ **or** $L[i] \leq R[j]$) **then**
4:         $B[k] \leftarrow L[i]$; $i \leftarrow i+1$  ▷ ties take $L$ first $\Rightarrow$ stability
5:     **else**
6:         $B[k] \leftarrow R[j]$; $j \leftarrow j+1$
7:     **end if**
8: **end for**
9: **return** $B$

## CORRECTNESS

**Theorem 1** (Permutation). *Algorithm 2 outputs a permutation of its inputs; hence Algorithm 1 outputs a permutation of $A$.*

*Sketch.* Each element of $L$ and $R$ is written exactly once to $B$; no new elements are created and none are omitted. By induction on recursion depth, the claim holds for the full algorithm. □

**Theorem 2** (Sortedness). *The output of Algorithm 2 is non-decreasing; thus, by induction on subproblem size,* MERGE-SORT *is sorted.*

*Sketch.* At every step, the minimum of the unmerged prefixes is appended. Since $L$ and $R$ are individually sorted, each choice preserves global order. Induction over $k$ establishes equation 1. □

**Theorem 3** (Stability). *If equal keys exist, their relative order from the input is preserved in the output.*

*Sketch.* On $L[i] = R[j]$, the merge rule selects $L[i]$ first, preserving left-to-right order of equal elements across subproblems. Within $L$ and $R$, stability holds by induction; thus equation 2. □

## COMPLEXITY

Let $T(n)$ denote worst-case time for size $n$. The recurrence $T(n) = 2\,T(n/2) + \Theta(n)$ solves to $T(n) \in \Theta(n \log n)$. Space is $S(n) = \Theta(n)$ to hold the merged buffer (with options for in-place variants).

## LIMITATIONS AND SCOPE

**Auxiliary memory.** The standard merge uses $\Theta(n)$ extra space; in-place variants reduce peak memory at the cost of implementation complexity and constant-factor overhead.
**Cache behavior.** Top-down recursion may be suboptimal for small subarrays; hybrid thresholds and bottom-up merging can improve locality.
**Parallel overhead.** Naive task spawning can exceed gains for small subproblems; work-stealing and grain-size control are required.
**Model assumptions.** Guarantees hold in the comparison model on total orders; partial orders or non-comparable keys require adaptations.

### A.3 DIVING DEEPER INTO THE PAPER

After the undergraduate student has read through their immersive introduction, they want to learn more about the details of MergeSort. To avoid repeating the core paper(Appendix A.2), we assume that the student has the core paper at hand as reference similar to a formula sheet and the AI agent will reference the relevant section while guiding them

through the paper. Please note that we only do that here to avoid repeating the components of the core paper; however, in practice, the AI agent will directly insert the relevant reference at the right place.

---

### ☰ The undergraduate student diving deeper

**Goal.** By the end of this ramp, you can read the formal core paper smoothly: you will know what a *total order* is, what *stability* means, how the *merge* step works, and why MergeSort runs in $\Theta(n \log n)$. This directly maps to *Model*, *Correctness*, and *Complexity* in the core paper as well as Algorithms 1–2.

**1) The world MergeSort lives in (vocabulary).**

- *Total order.* We sort items drawn from a set $X$ with a comparison $\leq$ so that for any $x, y \in X$ you can decide $x \leq y$ or $y \leq x$, and comparisons are consistent (transitive, antisymmetric).

- *Input/Output.* The input is an array (sequence) $A = \langle a_1, \ldots, a_n \rangle$. The output is another array $B = \langle b_1, \ldots, b_n \rangle$ containing the same elements, but in nondecreasing order.

- *Stability.* If two items are "equal" as keys, a stable sort keeps them in their original left-to-right order. In symbols: if $a_i = a_j$ with $i < j$, then in the output $B$ the copy that came from $a_i$ must appear before the one from $a_j$. In the core paper this is Equation (2) using the notation $\mathrm{pos}_B(\cdot)$.

**2) The big idea (divide & conquer).**

1. *Divide:* Split the array into two halves.
2. *Conquer:* Recursively sort each half.
3. *Combine:* **Merge** the two sorted halves into one sorted whole in linear time.

This is exactly what Algorithm 1 (MERGESORT) expresses.

**3) How the merge works (and why it's stable).** Assume you already have two sorted arrays:

$$L = \langle \ell_1, \ldots, \ell_\ell \rangle, \qquad R = \langle r_1, \ldots, r_r \rangle.$$

You repeatedly compare the *fronts* of $L$ and $R$ and move the smaller one into the output $B$. If they tie, you take from the left side first. That simple tie-break rule gives stability: equal keys preserve their left-to-right order across subproblems, which is exactly what Equation (2) requires.
*Key invariant while merging:* after $k-1$ placements, $B[1..k-1]$ is sorted and contains exactly the elements consumed from the prefixes of $L$ and $R$. Taking the smaller (or left-first on ties) preserves this invariant until $B$ is full. This is what Algorithm 2 (MERGE) formalizes line-by-line.

**4) A concrete walkthrough.** Consider $A = [5, 2, 4, 6, 1, 3]$.

1. Split repeatedly:

$$[5, 2, 4, 6, 1, 3]$$

$$\Rightarrow [5, 2, 4] \mid [6, 1, 3]$$

$$\Rightarrow [5, 2] \mid [4] \mid [6, 1] \mid [3].$$

$$\Rightarrow [5] \mid [2] \mid [4] \mid [6] \mid [1] \mid [3].$$

2. Base cases sort trivially.

3. Merge bottom-up:

$$[5] \bowtie [2] \to [2, 5], \quad [6] \bowtie [1] \to [1, 6],$$

$$[2, 5] \bowtie [4] \to [2, 4, 5],$$

$$[1, 6] \bowtie [3] \to [1, 3, 6],$$

$$[2, 4, 5] \bowtie [1, 3, 6] \to \boxed{[1, 2, 3, 4, 5, 6]}.$$

At every arrow, the invariant above ensures the partial output is already sorted.

**5) Why the result is sorted (Equation (1)).** Each merge always picks the smallest available next element from the two sorted fronts, so the output grows in nondecreasing order. By induction up the recursion tree, the final array satisfies $b_1 \leq b_2 \leq \cdots \leq b_n$, which is Equation (1) in the core paper.

**6) Why it's fast: the $\Theta(n \log n)$ intuition.** There are $\log_2 n$ levels of splitting (keep halving until size 1). Merging across any one level touches each element once, i.e., $\Theta(n)$ work per level. Hence total time is $\Theta(n)$ per level $\times \log n$ levels $= \Theta(n \log n)$, matching the formal recurrence $T(n) = 2T(n/2) + \Theta(n)$ solved in the core paper.

**7) Space and practical notes (reading the limitations).** Standard merging uses an auxiliary buffer of size $\Theta(n)$ to write the combined output cleanly. Variants exist that reduce peak extra space ("in-place merge") at the cost of more complex code and larger constants. These trade-offs are exactly what the *Limitations and Scope* paragraph flags for you.

**8) Checklist before you open the core paper page.**

- You can define total order, stability, and the merge invariant.

- You can explain why tie-breaking "take left on equality" yields stability.

- You can reproduce the $T(n) = 2T(n/2) + \Theta(n)$ reasoning and the $\log n$ levels idea.

*Now turn to the core paper (Appendix A.2):* match these ideas to the formal "Model" and "Correctness specification," then read Algorithm 1 (MERGESORT) and Algorithm 2 (MERGE) line-by-line with the invariant in mind.

---

### A.4 Bringing in Additional Context

While reading through the more detailed explanation, the student obtains a better sense of how the algorithm works. However, they do not understand the runtime complexity notation $\Theta(n \log n)$. So they ask the AI agent to expand on the runtime complexity:

---

### ☰ Additional context for the student

**Runtime complexity** is a way to describe how the number of steps an algorithm takes grows as the input size $n$ becomes larger. Instead of counting exact operations (which depends on the programming language, machine, etc.), we describe the *trend* of growth using mathematical notation.

**The symbols.**

- $O(f(n))$ (*big-O*) gives an *upper bound*: the algorithm never takes more than a constant times $f(n)$ steps (for large $n$).
  Example: saying $T(n) \in O(n^2)$ means the runtime grows at most quadratically in $n$.
- $\Omega(f(n))$ (*big-Omega*) gives a *lower bound*: the algorithm always takes at least a constant times $f(n)$ steps (for large $n$).

Example: $T(n) \in \Omega(n)$ means the runtime is at least proportional to $n$.

- $\Theta(f(n))$ (*big-Theta*) means both are true: the runtime is bounded above *and* below by $f(n)$ up to constant factors.
  Example: $T(n) \in \Theta(n \log n)$ means the runtime grows on the order of $n \log n$, neither faster nor slower (ignoring constants).

**Why this matters.** Runtime complexity abstracts away details like exact machine speed and instead focuses on how quickly the difficulty of the problem grows as inputs get bigger. This is why MergeSort's $\Theta(n \log n)$ tells us it is asymptotically more efficient than algorithms like Insertion-Sort, which is $\Theta(n^2)$.

---

## A.5 OFF TO NEW ADVENTURES

Now equipped with the knowledge of analyzing runtime complexities, the student remembers a sorting algorithm they wrote during an internship to clean up log files. Back then, different systems wrote logs into the same log file; thus, it can happen that the logs are not in order due to clock skews or late write events. The goal was to sort each line according to the corresponding timestamp. The algorithm of the student was able to sort the list with just a single pass through, and just swapping subsequent lines in case they are out of order. Excited by this discovery, the student shows the Python code to the AI agent and claims that this algorithm clearly runs in $O(n)$.

---

### ☰ An AI agent helps the student to sail new adventures

That is a clever approach — what you implemented is essentially a version of *BubbleSort* tailored to log files. And you are right: on logs that are *almost sorted*, it can run very quickly. If every timestamp is already in the correct place (or just one step away), a single left-to-right pass is enough, so the runtime is $O(n)$. This matches your observation from practice.
But here's the important distinction: **runtime complexity is about the worst case as well**. If the log lines are badly out of order — for example, if the file is completely reversed — then each pass only moves items one step closer to their correct place. In that case, the algorithm requires about $n$ passes over $n$ items, which is $O(n^2)$ time.

So your algorithm is well-suited for log files that are *nearly sorted* (a common situation when only occasional clock skew or late writes occur). But as a general-purpose sorting algorithm, it does not guarantee $O(n)$ performance. That is why algorithms like MergeSort, with a reliable $\Theta(n \log n)$ runtime for *every* input, are preferred in practice.
**Takeaway:** The same algorithm can look linear on "easy" inputs but quadratic on "hard" ones. That's why computer scientists carefully distinguish *best-case*, *average-case*, and *worst-case* complexities.

---

## A.6 DYNAMICALLY ADAPTING NOTATION

In the meantime, also the professor was able to work their way through the paper. However, the professor has a hard time understanding Alg. 2. The professor is a functional programmer at heart, and did a lot of pattern matching in the past. Thus, the professor asks the AI agent to translate it to a notation the professor is more familiar with.

---

### ☰ MergeSort algorithm tailored to the background of the professor

We work with lists built from either the empty list $[\,]$ or a cons cell $x{:}xs$ (element $x$ followed by list $xs$).

**Stable Merge.**

$$\mathrm{merge}([\,],\ ys) = ys, \qquad \mathrm{merge}(xs,\ [\,]) = xs,$$

$$\mathrm{merge}(x{:}xs,\ y{:}ys) = \begin{cases} x{:}\mathrm{merge}(xs,\ y{:}ys), & \text{if } x \leq y, \\ y{:}\mathrm{merge}(x{:}xs,\ ys), & \text{otherwise.} \end{cases}$$

*Explanation.* If one input is empty, return the other. Otherwise compare the two heads $x$ and $y$:

- If $x \leq y$, output $x$ and continue with the rest of $xs$ and the full $y : ys$.
- If $y < x$, output $y$ and continue with the full $x : xs$ and the rest of $ys$.

On ties we choose $x$ (the left head) first, which enforces **stability**.

---

For the AI agent to be able to generate personalized notation, all variables need to be clearly defined, if not clear from the context. Please note that this goes far beyond translating pseudo code. Some research labs denote matrices and vectors in bold letters $\mathbf{A} \in \mathbb{R}^{n \times m}$, while scalars are denoted using normal font $a \in \mathbb{R}$. Indexing can be done simply in the subscript $A_{i,j}$, with parentheses $A(i,j)$, or a mixture of both $A_{(i,j)}$. Sets might be denoted with blackboard bold font $\mathbb{N}$, but also calligraphic fonts $\mathcal{N}$ are common. The latter is also often used to denote a normal distribution $x \sim \mathcal{N}(0,1)$. Some researchers also introduce abbreviations to save space, which unfamiliar researchers might find difficult to read. All these conventions can be tailored to the reader such that the focus is on understanding the content of the paper rather than deciphering the notation.

## B    ETHICS STATEMENT

Our proposed paradigm shift raises two central ethical concerns. First, while personalized contextualization could democratize research by adapting to diverse educational backgrounds, it risks creating unequal knowledge experiences based on AI system quality and availability. Readers with limited access to sophisticated AI tools may face systematic disadvantages. Second, AI systems may introduce or amplify biases in how research is presented to different audiences, potentially steering interpretation based on reader demographics or prior beliefs rather than scientific merit. The opacity of these personalization decisions could hide systematic distortions in scientific communication.

Mitigation strategies could be to enforce that immersive reception tools are highly accessible and offer comprehensive subsidized access for disadvantaged regions and individuals. User profiles should not contain demographic information, and it should be ensured that the AI does not infer it. The establishment of ethics boards and regular reviews based on ethical considerations could serve as additional safeguards.

## C    STATEMENT ON THE USE OF LLMS

LLMs have been used for generating the examples in Appendix A, as described. In the creation of this manuscript, LLMs have been used for literature search and editorial purposes.

