# OpenReview forum: "Position: We Should Read and Write Papers Differently in the Age of AI"
_ICLR.cc/2026/Conference — ICLR 2026 Conference Desk Rejected Submission_

### Official Review · Reviewer_eTX1 · 2025-10-20

**Soundness:** 1
**Presentation:** 1
**Contribution:** 1
**Rating:** 0
**Confidence:** 5

**Summary:**

This paper changed the page margin, which is not allowed by ICLR.

**Strengths:**

This paper changed the page margin, which is not allowed by ICLR.

**Weaknesses:**

This paper changed the page margin, which is not allowed by ICLR.

**Questions:**

This paper changed the page margin, which is not allowed by ICLR.

---

### Official Review · Reviewer_pnHY · 2025-10-30

**Soundness:** 2
**Presentation:** 3
**Contribution:** 2
**Rating:** 2
**Confidence:** 4

**Summary:**

The paper argues that papers should be read and written differently in the age of AI. The reason is that the paper interaction will be mediated ubiquitously by AI. This will allow 1) writing to be made more efficient, 2) author-induced biases to be reduced, and 3) make paper reception more efficient.

Instead of the paper being provided by the author, it can be generated for the reader.

Some standard parts of a paper, such as context, motivation, background, and related work will be generated for the reader by an AI, while the other parts (the core paper) will be written by the authors, such as novel contributions, methodology formalization of the methodology and the evaluation.

Researchers can put less time into writing papers, as less paper content translates into less effort.

**Strengths:**

### Strengths
* Clearly, scrutinizing the way we do science to improve it is good. The format of the scientific paper is still exactly the same as it has been for a long time – if not the start of science. Many have questioned whether a paper structured in this way is the best way to communicate research.
* The paper is well-written, and I like its structure. I especially like how the authors present the assumptions that they operate under and the alternate views section. Nice!
* Living survey papers is a good idea that could be managed by the community and AI generated (to some degree). While the idea is not new, the potential with AI is interesting. I would have loved to read a paper that dug into this idea.

**Weaknesses:**

### Weaknesses

* The assumption that AI will mediate all research consumed by humans is a very strong assumption. It means that everyone that wants to consume research needs access to AI. Also, these AIs must be personalized for individual researchers to communicate research to its users. This introduces a huge complexity in research communication compared to the simplicity of PDF documents. It requires a huge infrastructure that all researchers must have access to in order to conduct research. Poor countries, communities, and researchers will have even less opportunity to do research, which will further drive inequality in the world – both at a national and a personal level.
* Will the plurality of AI vendors ensure partiality? One argument is that no vendor will decrease the output quality for the sake of the agenda. I find this argument a bit naïve given how we see that the world is going the exact opposite way the last few years. Commercial actors are very prone to change their policies according to the government’s changing views. The other argument is that the plurality of tools will cancel out overall. However, given that some tools are more prevalent in some parts of the world than others, there could very well be silos that do not exchange worldviews and scientific views freely.
* More importantly, the paper argues that author partiality is a poor thing that hampers the progress of science. This partiality is distributed and has a lot of noise. However, because it is random over so many authors it could be thought of as a form of regularization process, while a systemic bias from a few AI agents could be thought of as having less regularization effect and therefore hamper the speed of progress.
* The readers’ needs can be fully supported in the current setup, especially if the assumption is that AI will get better, which is an assumption made explicit in the paper. AI will be able to provide a context that builds upon what is written in the paper, and it can be personalized to the reader. Hence, the proposed solution will not necessarily help the reader more than the current alternative, which is to rely on PDFs and own AIs.
* An assumption is that it is better for the author to write less and to not focus on background and putting the work into context. I am not sure this is a good assumption, as the author should ground the research into context and explain why this research makes sense. This helps guide research, and these motivations should be made explicitly by the authors. AIs should not guess about this.
* An issue that researchers must manage today is the huge growth in research output. It seems to me that this solution will not necessarily help to mitigate this issue but instead fuels it. More research does not necessarily mean better research.
* The position that is proposed here is that the paper of the future is only text. IS this a good idea? Would this help reproducibility? Would it not make sense to make the paper of the future rely on more than text, such as data and code as well, as discussed by Gil et al. (2016) and  Gundersen (2021), especially in the computer sciences? See also Gil (2021).
* AI will only get better, so why not make AI support the current workflow? It requires less change, which is hard for humans and communities to accommodate, less infrastructure, and it keeps research output simple (documents, code and data).
* I do not see how the proposed solution will reduce knowledge barriers when compared to PDF with a personalized AI. Why would not this solution reduce the knowledge barriers to the same degree? I do not see any reasons that PDF (text) and personalized AI would not have the same immersive reception as the proposed solution without relying on a worldwide change in research infrastructure and practices.
* The changes that is proposed are not analyzed considering the scientific enterprise as a whole. The paper and its structure alone are mainly analyzed -- with one exception: citations. The scientific enterprise is to a large degree centered around scientific articles – for better or worse. This means that changes to how papers are conceptually understood changes the whole scientific endeavor to some degree. The analysis is therefore too narrow because the implications of the proposed changes are large.
* One thing is forgotten in the temporal dimension, which is that a paper documents the current understanding and the state-of-the-art at a point in time. According to the paper, this is an issue. However, explaining the context that the paper actually was written in (whether it is a context that reflects the “true” world or not) is important. If the context changes with the reader and time, many papers will not make sense – even though the context they were written in made sense, ref. discussions about ether before the relativity theory. Therefore, getting the context of string theory and quantum mechanics when reading about ether will not help one understand the research on ether.
* To some degree, the position paper reflects a book-of-truth attitude towards the literature, Davis-Strober er al. (2025), meaning that the literature should reflect knowledge rather than a book-of-dialogues where papers might not reflect truths, but ideas, and that papers are dialogues between researchers in a search for truth. This is reflected in how state-of-the-art is interpreted as something crisp that can be documented in living surveys. Surveys (good ones at least) tries to put the literature in a context and structure it. Therefore, surveys are views or framings on the state-of-the-art rather than a true reflection of knowledge. Hence, there is not only one views of it, but many.
* Will the paper reception be more efficient when context between producer (author) and consumer (reader) is different?
* Cornserstone papers will go through an extensive review process. What about the other papers? Should they not be extensively reviewed? How will this improve the quality of science?
* Not clear to me how author-induced biases will be reduced. How will this solve it? Will the proposed paper structure hide the context it was written in? Will the context of the author be perfectly described by an AI? How can we be sure about this?  There need to be a connection to the literature, and it needs to be made explicit. Does it not?
* I do not understand how citations scores would be computed given that the proposal relies on commercial AIs that are personalized. How should citation scores be shared? The current solution is computable from the set of all papers, which admittedly is an effort that requires huge resources. However, no extra information than the papers themselves is required to make a universal score that reflects the true citation score. The proposed solution seems unwieldly and impossible to manage unless all vendors of AIs collaborated. If citations are based on what AIs decide when they present related research to the reader of a paper, do they not control which papers that are shown? Will not this introduce strong biases? An AI deciding to present the result of a paper does not mean that the reader found that paper valuable. The good thing about the current system is that a citation is evidence the author found the paper useful to some degree – enough to cite. What am I missing here?
* It is stated that on line 320 “In practice, many authors already treat introductions and related work as formulaic, often …”. How do you know this? Is this statement representative? Is this statement just biased by the authors views? Same for the statement on line 337 “… ana acceleration of the scientific progress would decrease its environmental impact in the long term”. What is the evidence for this statement?
* The objective of having introduction and related work parts of research articles is not only to give the reader an overview of relevant research and background, but also to frame the work in its time and in the view of the authors. Scientists do not always agree on what is important and what is not. Scientists do not even agree that there exists absolute truth, and they approach research with different attitudes toward the body of literature. Some could think of it as dialogues between scientists while others would consider it a book of truths, see for example Davis-Stober et al. (2025) for a discussion.

### References
Davis-Stober, C. P., Sarafoglou, A., Aczel, B., Chandramouli, S. H., Errington, T. M., Field, S. M., ... & Gundersen, O. E. (2025). How can we make sound replication decisions?. Proceedings of the National Academy of Sciences, 122(5), e2401236121.

Gil, Y., David, C. H., Demir, I., Essawy, B. T., Fulweiler, R. W., Goodall, J. L., ... & Yu, X. (2016). Toward the Geoscience Paper of the Future: Best practices for documenting and sharing research from data to software to provenance. Earth and Space Science, 3(10), 388-415.

Gil, Y. (2022). Will AI write scientific papers in the future?. AI Magazine, 42(4), 3-15.

Gundersen, O. E. (2021). The fundamental principles of reproducibility. Philosophical Transactions of the Royal Society A, 379(2197), 20200210.

**Questions:**

See above.

---

### Official Review · Reviewer_MNUL · 2025-10-31

**Soundness:** 2
**Presentation:** 2
**Contribution:** 1
**Rating:** 2
**Confidence:** 4

**Summary:**

This position paper argues that the traditional, author-driven structure of scientific publications is no longer well suited to the AI era. It promotes a reader-centered approach in which papers focus on their essential technical contributions, while contextual sections such as the introduction and related work are generated dynamically by AI to match each reader’s background and needs. The authors discuss possible implications for reproducibility, citation practices, and survey literature, introducing concepts like AI-generated state-of-the-field reports and rigorously reviewed cornerstone papers. Although the paper acknowledges challenges including hallucination, energy consumption, and long-term archiving, it presents an ambitious and timely vision for a more adaptive and personalized model of scientific communication.

**Strengths:**

The paper addresses a highly relevant and timely question of how LLMs or AI may transform scientific communication. Its vision is forward-looking and engages meaningfully with an emerging challenge. The authors present a well-structured and coherent argument for shifting from an author-centered to a reader-centered model, supported by clear examples and some explainations. Moreover, the proposal for dual submission formats and new mechanisms, such as cornerstone papers and AI-generated sota reports, offers concrete and practical steps toward realizing this vision rather than leaving it at a purely conceptual level.

**Weaknesses:**

Overall, I think this paper is far from accepted by top AI conference. The assumption and proposed new system is questionable, specifically:

1. The paper assumes that AI systems will be objective and continuously improving, but it underestimates risks such as hallucinations, bias from training data, and potential corporate or institutional influence.

2. While the vision is compelling, the paper provides few details on how the proposed AI-mediated publication infrastructure would be developed, standardized, or governed in practice.

3. Reducing papers to “core content” may weaken the reflective and integrative aspects of scientific writing, which are often essential for conceptual clarity and creativity.

4. The proposal is mainly illustrated through computer science examples, with limited consideration of fields where contextual argumentation and narrative reasoning are central. I'm not quite sure about the LLMs' knowledge accuracy in other domains, thus I might concern that cross-domain knowledge is still too early to implement such new framework. At least, in this paper, the authors should prove current LLMs or AI can handle cross-domain generation acceptablely.

5. The environmental cost of LLMs usage is acknowledged but treated superficially without quantitative analysis or mitigation strategies.

6. The paper is not well organized, no figure to explain their idea, no data or statistics and any investigation to support their claims, which make these points not convincing.

**Questions:**

1. In line 62-63, 'we base our core argumentation mainly on what we deem possible with the tools available today'. What evidence or literature supports this claim? No citation or empirical reference is provided, which makes the statement speculative rather than scientific.

2. Some references are not new and necessary, some paragraph are in very bad logical way, for example, in line 81-85, why these two citations are necessary? This paragraph is also in bad logic.

3. In line 179-184, how would the system determine a reader’s expertise level and information needs? Has any prior research demonstrated feasibility?

---

### Official Review · Reviewer_epag · 2025-11-01

**Soundness:** 1
**Presentation:** 1
**Contribution:** 1
**Rating:** 0
**Confidence:** 5

**Summary:**

This is a blue sky paper arguing for radically changing the scientific communication process, so that authors need not write a large part of the paper framing they do today; instead, AI should generate that type of content at reading time. The paper details potential benefits and risks, and describes mechanisms for addressing potential problems, like creating citation counts that are informed by text generated by readers' AI systems.

**Strengths:**

The idea proposed is interesting, and I agree that the scientific communication process is under strain---and that AI adoption by readers will play an increasingly large role in how scientists consume information. I appreciate that the paper walks through assumptions, risks, etc. systematically and provides arguments. The reader-derived citation idea is also interesting and feels fairly novel though I am aware of contemporaneous work making a similar argument (https://allenai.org/blog/asta-citations).

**Weaknesses:**

Aside from questionable relevance to the CFP (I do not see a call for blue sky papers https://iclr.cc/Conferences/2026/CallForPapers), this paper does not engage to a sufficient extent with the prior literature on this topic, instead presenting the idea as largely novel. However, aside from the contemporaneous work above, there are numerous proposals that have been written on how to revise peer review (Ko. 2023; Lecun. 2013), how to reduce the size of publications (e.g., Micropublication Biology), or leverage AI for reading support (Lo et al. 2024). This paper would also be stronger if it were supported by evidence from a proof of concept demonstration, as people have done with other forward-looking ideas like micropublications or even a fully AI written and reviewed conference (https://openreview.net/group?id=Agents4Science/2025/Conference). The writing style of the paper is also at times overly casual e.g., with headings like "Off to new adventures."

**Questions:**

- How does this work differ from prior related work as described above?
- What evidence is there to support the ideas proposed here? Is there any evidence or learnings from a proof of concept demonstration?

---

### Note · Program_Chairs · 2026-01-17
**Submission Desk Rejected by Program Chairs**

The following references in this submission do not refer to real documents and/or have major errors in bibliographic information:

 Michael Kacena et al. Artificial intelligence in medical writing: Promises and pitfalls. The Lancet Digital Health, 6 (2):e85-e87, 2024b.